# The Interaction Effect of Anti-RgpA and Anti-PPAD Antibody Titers: An Indicator for Rheumatoid Arthritis Diagnosis

**DOI:** 10.3390/jcm12083027

**Published:** 2023-04-21

**Authors:** Diana Marcela Castillo, Gloria Inés Lafaurie, Consuelo Romero-Sánchez, Nathaly Andrea Delgadillo, Yormaris Castillo, Wilson Bautista-Molano, César Pacheco-Tena, Juan Manuel Bello-Gualtero, Philippe Chalem-Choueka, Jaime E. Castellanos

**Affiliations:** 1Unidad de Investigación Básica Oral-UIBO, Vicerrectoría de Investigaciones, Facultad de Odontología, Universidad El Bosque, Bogotá 110121, Colombia; lafauriegloria@unbosque.edu.co (G.I.L.); ndelgadillos@unbosque.edu.co (N.A.D.); castilloyormaris@unbosque.edu.co (Y.C.); 2Cellular and Molecular Immunilogy Group (INMUBO), School of Dentistry, Universidad El Bosque, Bogotá 110121, Colombia; romeromaria@unbosque.edu.co (C.R.-S.); wbautistam@unbosque.edu.co (W.B.-M.); 3Clinical Immunology Group, Rheumatology and Immunology Department, Hospital Militar Central, Bogotá 110231, Colombia; juanmabello36@gmail.com; 4Clinical Immunology Group, School of Medicine, Universidad Militar Nueva Granada, Bogotá 110231, Colombia; 5Investigación Y Biomedicina De Chihuahua S.C., Chihuahua 31205, Mexico; dr.cesarpacheco@gmail.com; 6Fundación Instituto Fernando Chalem, Bogotá 111211, Colombia; p_chalem@yahoo.com; 7Facultad de Odontología, Universidad Nacional de Colombia, Bogotá 111321, Colombia; jecastellanosp@unal.edu.co

**Keywords:** gingipains, PPAD, rheumatoid arthritis, periodontitis, RgpA

## Abstract

*Porphyromonas gingivalis* secretes virulence factors like Arg-gingipains and peptidyl arginine deiminase (PPAD), that are associated with rheumatoid arthritis (RA) pathogenesis. However, there is no information regarding the antibody titers for these bacterial enzymes as systemic indicators or biomarkers in RA. In this cross-sectional study, 255 individuals were evaluated: 143 were diagnosed with RA, and 112 were without RA. Logistic regression models adjusted for age, sex, basal metabolic index, smoking, and periodontitis severity were used to evaluate the association of RA with rheumatoid factor (RF), anti-citrullinated protein antibodies (ACPAs), erythrocyte sedimentation rate, high sensitivity C-reactive protein, anti-RgpA, anti-PPAD, and double positive anti-RgpA/anti-PPAD. It was found that RF (odds ratio [OR] 10.6; 95% confidence interval [CI] 4.4–25), ACPAs (OR 13.7; 95% CI 5.1–35), and anti-RgpA/anti-PPAD double positivity (OR 6.63; 95% CI 1.61–27) were associated with RA diagnoses. Anti-RgpA was also associated with RA (OR 4.09; 95% CI 1.2–13.9). The combination of anti-RgpA/anti-PPAD showed a high specificity of 93.7% and 82.5% PPV in identifying individuals with RA. RgpA antibodies were associated with the periodontal inflammatory index in RA individuals (*p* < 0.05). The double positivity of the anti-RgpA/anti-PPAD antibodies enhanced the diagnosis of RA. Therefore, RgpA antibodies and anti-RgpA/anti-PPAD may be biomarkers for RA.

## 1. Introduction

Rheumatoid arthritis (RA) is a chronic, progressive inflammatory disorder that presents as asymmetric polyarthritis of small and large joints due to systemic inflammation [1]. A specific characteristic of RA is the presence of high quantities of autoantibodies against post-translationally modified proteins in synovial tissue fluids [2]. Citrullination, which plays an essential role in promoting autoimmunity, is a post-translational modification of proteins, including modifications of the amino acid side chain leading to the conversion of peptidyl-arginine to peptidyl-citrulline in RA and aberrant protein folding [3]. Protein citrullination is catalyzed by the peptidyl arginine deiminase (PAD) enzyme family, linked with the occurrence of autoimmune-mediated inflammation by its capacity to elicit inflammatory responses such as immune cell differentiation and immune response [4,5]. PAD-rich macrophages and monocytes are degraded by apoptosis and citrullinated extracellular proteins such as vimentin, α-enolase, and fibrinogen in the joints, generating a loss of immune tolerance [6]. Anti-citrullinated protein antibodies (ACPAs) stimulate the production of proinflammatory cytokines, osteoclastogenesis, and neutrophil extracellular trap formation [7].

*Porphyromonas gingivalis* is a keystone pathogen associated with the progression of chronic periodontitis by secretion of virulence factors, including the gingipains, which are encoded by three genes (*rgpA, rgpB*, and *kgp*). These are known as gingipain-R (RgpA and RgpB) and gingipain-K (Kgp) according to their hydrolysis specificity for Arg-Xaa or Lys-Xaa bonds, respectively [8] and peptidyl arginine deiminase of *P. gingivalis* (PPAD) in the periodontal pocket [9]. PPAD requires a higher pH for its activity at the carboxyterminal of proteins and the free arginine residues cleaved by gingipains to generate autoantigens derived from fibrinogen and α-enolase diffuse to distant tissues through outer membrane vesicles (OMVs) or as a soluble enzyme [10,11,12].

Periodontitis has been recognized as an independent factor associated with RA, particularly in individuals with more than 5 years of disease [13], and antibodies against *P. gingivalis* are significantly associated with RA in multiple studies [14]. However, increased levels of antibodies anti-*P. gingivalis* have also been found to be significantly associated with RA [15,16] and early RA [17] in individuals with periodontitis.

Currently, there is still controversy regarding the importance of PPAD in the citrullination process in RA. Although it has been hypothesized that PPAD could contribute to breaking immune tolerance and can generate antibodies [16,18,19], others did not find that PPAD auto-citrullination is the mechanism that links periodontitis and RA [20,21]. The concentration of anti-RgpB antibodies in established RA has been controversial [16,18]. However, RgpA gingipain has not been evaluated, and it has shown more significant activity and virulence because it can be found in a soluble form and its complete hemagglutinin adhesin domain together with the catalytic domain (HRgpA) or incomplete (RgpAcat), are highly homologous to RgpBcat [8]. Additionally, two non-soluble forms are associated with the membrane of the bacteria and is only present in OMVs as mt-RgpAcat and mt-HRgpA [22]. Since PPAD activity is directly related to RgpA gingipain activity, it is critical to evaluate its association and the impact of the PPAD/RgpA interaction with RA. This study aimed to assess the association of anti-RgpA and anti-PPAD antibodies and the interaction of anti-RgpA/anti-PPAD antibodies in individuals with RA and to analyze their sensitivity and specificity to detect RA individuals.

## 2. Materials and Methods

### 2.1. Type of Study

This cross-sectional observational study was approved by the Ethics Committee of the Universidad El Bosque (012-2016) and Hospital Militar Central (HMC-2016-018) and followed the principles of the Helsinki Declaration on Human Experimentation.

### 2.2. Population and Sample

255 individuals were evaluated: 143 diagnosed with RA, according to the American College of Rheumatology/European League Against Rheumatism 2010 classification criteria [23], and 112 individuals without RA. The sample size was established to detect a difference between the highest titers of anti-RgpA (quartile 1; greater than 75th percentile) in patients with RA (32%) and the control group (12%) with an odds ratio (OR) ≥ 3.45, according to a pilot study that evaluated 50% of the final sample for a minimum sample of 102 individuals per group. Individuals with RA were recruited from the Hospital Militar and Fundación Instituto Fernando Chalem, and the control group from the Hospital Militar. All patients included in this study signed informed consent, approved by the ethics committee. In the RA and control groups, a rheumatologist applied a questionnaire that included questions about age, sex, height, weight, body mass index (BMI), smoking habits, diagnosis, type of treatment, comorbidities, and swollen and painful joints, among others.

### 2.3. Inclusion Criteria for RA Individuals and Controls

Figure 1 describes the selection of the population included in this study. Individuals aged between 35 and 65 years with an RA diagnosis established for a minimum of two years were selected for the RA group and those with no RA signs were selected for the control group.

### 2.4. Exclusion Criteria

The following individuals were excluded: those with ongoing infectious processes or diagnoses of neoplasms, diabetes mellitus, autoimmune or systemic diseases; lactating women; those who had undergone antibiotic treatment in the last 90 days before sample collection; those who had orthodontic devices; and those who had undergone periodontal therapy in the past 6 months.

### 2.5. Evaluation of Clinical Periodontal Status

All individuals were evaluated by two calibrated periodontists who examined their periodontal status, which was evaluated in the full mouth using a periodontal probe (Hu-Friedy Mfg Inc.-Qulix™) to determine the following indices: plaque index (PI), gingival index (GI), bleeding on probing (BOP), pocket depth (PD), and clinical attachment loss (CAL) in millimeters. Calibration for periodontal indices included BOP (with an inter-examiner correlation coefficient [IE-ICC] 0.88–0.90), PI ([IE-ICC] 0.94–0.98), GI ([IE-ICC] 0.88–0.90), probing depth ([IE-ICC] 0.96–0.98), and CAL ([IE-ICC] 0.90–0.96). Individuals were classified as having periodontitis if they had at least two or more teeth with proximal CAL of at least 2 mm with a pocket depth of at least 4 mm and classified as low, moderate, or severe according to the Centre for Disease Control and Prevention/American Academy of Periodontology index [24,25]. Individuals without periodontitis were categorized as healthy or with gingivitis. Additionally, we collected subgingival plaque samples from the deepest sites of the gingival sulcus in patients with periodontitis and one site per sextant in control patients. The supragingival plaque was removed using a sterile curette, and the sample site was isolated using cotton rolls. Samples were taken from six selected sites, one in each quadrant. However, in patients without periodontitis, samples were taken from the mesiobuccal sites of the first molars and, when absent, from the adjacent second molars. Samples were obtained from two consecutively inserted sterile paper points per site, which were left in place at the bottom of the sulcus/pocket for 10 s with paper points (NewStetic^®^) to detect *P. gingivalis* using quantitative polymerase chain reaction (qPCR).

### 2.6. Identification and Quantification of P. gingivalis in Subgingival Plaque

The qPCR was performed following the recommendations of Boutaga et al. [26]. Primers and probes were used for the gene that codes for 16SrRNA, as has been previously standardized in our laboratory [17,26].

### 2.7. Measurement of Arthritis Rheumatic Biomarkers

A high-sensitivity C-reactive protein (hsCRP) test was performed using chemiluminescence (Immulite 1000-Siemmens^®^) and a value of 3 mg/L was considered positive. The erythrocyte sedimentation rate (ESR) was measured using the photometric method (Ali-Fax-Test 1 THL Ali-FAX^®^Polverara (PD)-Italy) and values greater than 20 mm were considered elevated. A sandwich-type enzyme-linked immunosorbent assay (ELISA) system was used to measure antibodies to ACPA-IgG/IgA (Quanta lite^®^CCP3.1 IgG/IgA, INNOVA-Diagnosis) in serum. The results were expressed in ELISA units/mL, where results greater than 20IU were considered positive. Rheumatoid factor (RF) was quantified using nephelometry (Beckman Coulter, Immage800^®^), and values greater than 20 U were deemed positive. With the hsCRP and the data of the rheumatological clinical evaluation, the DAS-28 score was extracted to determine the activity of the disease only for the group with a diagnosis of RA because this rate of disease is not measured in healthy individuals [27].

### 2.8. Detection of Anti-RgpA and Anti-PPAD Antibodies by ELISA

#### 2.8.1. Purification of Native RgpA

RgpA protein was purified from OMVs of *P. gingivalis* ATCC33277 following a previously described protocol [28,29]. Once the OMVs were obtained, they were sonicated (Vibra-cell VCX130-Sonics) and ion-exchange liquid-phase chromatography (FPLC) (BioLogic DuoFlow™ BioRad) was performed. An initial cycle of anion exchange chromatography allowed for the collection of bioactive fractions, which were separated in the second cycle of cation exchange chromatography, followed by filtration using a 3 kDa filter (Pall Nanosep™). Protein purity was confirmed using sodium dodecyl-sulfate polyacrylamide gel electrophoresis (SDS-PAGE) and Western blot [30].

#### 2.8.2. Recombinant PPAD Production

The amino and carboxy terminals of PPAD were produced recombinantly. In brief, DNA from *P. gingivalis* ATCC33277 was independently amplified using PCR to obtain the corresponding fragments which were cloned in the plasmid pGEM-RT (Promega) and propagated in *Escherichia coli* Top10 in Luria Bertani (LB) agar (Difco™) and selected with ampicillin (Sigma) 100 µg/mL. PCR confirmed the positive colonies, and the fragment of interest was released using Xbal (Promega) and Nsil (Thermo Fisher-Scientific) restriction enzymes. Later, these were subcloned into the expression plasmid (Champion™ pET302/NT-His and pET303/CT. Invitrogen) and transformed into *E. coli* BL21. The inserts were verified by PCR, and the directionality of the inserts was confirmed by PCR and Sanger sequencing. Expression of PPAD-NT and PPAD-CT was induced with 1 mM isopropyl-β-D-thiogalactopyranoside (IPTG) (Pomega). The protein was purified using the Ni-NTA resin protein purification system (HisPur™ Ni-NTA Thermo) following the manufacturer’s recommendations.

#### 2.8.3. Indirect ELISA

An indirect ELISA was standardized in a 96-well flat-bottom polyethylene plate (Greiner Bio-One) which was coated with 20 µg/mL of purified RgpA for the determination of antibodies against the RgpA protein. Another 96-well plate was coated with 20 µg/mL of PPAD-CT and PPAD-NT to determine the levels of anti-PPAD antibodies. Both proteins were diluted in carbonate–bicarbonate buffer (pH 9.6) and incubated overnight at 4 °C, followed by three washes (Thermo-Scientific™ Wellwash™) with phosphate-buffered saline (PBS)-Tween (0.1%, pH 7.5). Nonspecific sites were blocked (StabilGuard^®^, SurModics) for 1 h at 18–20 °C and washed. Serum from each patient was added at a 1:400 dilution prepared in PBS-Tween (0.1%, pH 7.4)-3% skimmed milk and incubated for 2 h at 37 °C. After washing, biotinylated anti-human IgG (antibody polyclonal rabbit anti-human IgG HRP-Dako) diluted 1:10,000 was added and incubated for 30 min at 37 °C. The antigen–antibody reaction was developed using H_2_O_2_ and tetramethylbenzidine peroxidase substrate (TMB) (SERA CARE-KPL50-76-03), for 10 min, and the reaction was stopped using 1 M orthophosphoric acid (H_3_PO_4_) (Emsure^®^1000. Merck-Sigma-Aldrich). Optical density (OD) was read at 450 nm in an ELISA reader (Infinite^®^ 200PRO-Tecan). The positive serum sample came from a patient with periodontitis in whom the presence of *P. gingivalis* could be detected in peripheral blood after periodontal treatment by blood culture. Additionally, this serum contained *P. gingivalis* in subgingival plaques and anti-*P. gingivalis* antibodies (OD > 0.1). A negative serum control was obtained from a volunteer with no pockets and was negative for *P. gingivalis* in subgingival plaques and who did not present anti-*P. gingivalis* or RA markers (OD < 0.005). OD measurements of anti-RgpA and anti-PPAD antibodies were grouped into quartiles based on the median, defining the groups according to the titers: Q1 (values greater than 75th percentile), Q2 (values greater than 50th percentile), and Q3–Q4 (values below 50th percentile). Statistical differences between anti-RgpA and anti-PPAD titers (Q1/Q1, Q1/Q2) were calculated using STATA software v11.

### 2.9. Statistical Analysis

The Kolmogórov–Smirnov test was used to evaluate the distribution of continuous data, expressed as median and interquartile ranges. Absolute and relative frequencies were used to estimate the categorical variables. Clinical and sociodemographic data were calculated based on RA status and compared using the Mann–Whitney U-test, Chi-square, or Fisher test. All variables that showed significant associations with RA (*p* < 0.10) in the bivariate analysis were included in a multivariate logistic regression analysis to establish the OR at 95% confidence intervals (CIs). The primary outcome was RA diagnosis, and the independent variables were anti-RgpA, anti-PPAD, anti-RgpA-anti-PPAD double positive and RA markers (RF, ACPAs, ESR, and hsPCR). Three models were used to establish the association between anti-RgpA, anti-RgpA-PPAD double positive, and RA systemic markers. Model 1, Model 2, and Model 3 included anti-RgpA (Q1 and Q2), anti-PPAD (Q1 and Q2), and anti-RgpA, anti-PPAD, and anti-RgpA-anti-PPAD double positive (Q1/Q1, Q1/Q2), respectively. All models were adjusted for age, sex, BMI, smoking habits (SH), and severity of periodontitis. The adjusted and unadjusted models were verified by the goodness-of-fit test and compared using the likelihood-ratio Chi-square test (G2) and the Bayesian Information Criterion [31]. All data analysis was performed using STATA-v11.

To establish the operational ability of anti-RgpA, anti-PPAD, and RgpA-PPAD diagnostic tests, sensitivity, specificity, predictive positive value (PPV), predictive negative value (PNV), and area under the curve were obtained for each quartile and their combinations. Additionally, bivariate analysis was performed to establish associations between anti-RgpA, anti-PPAD, and anti-RgpA-anti-PPAD double positive and RA systemic markers (ESR, RF, ACPAs, CRP). Anti-RgpA and anti-PPAD were compared between RA and controls according to periodontal indices including, PI, GI, BOP, PD, and CAL.

## 3. Results

Although age and sex data were significantly different between the RA and control groups, no differences were found in BMI or SH (Table 1). The values for RA markers, such as ACPAs, ESR, hsCRP, and RF, were higher in the patient group (*p* < 0.05). Most of the patients had >5 years of evolution of RA and were treated with conventional DMARDs and only a small percentage of patients (4.19%) were considered to have high disease activity (Table 1). Periodontal and microbiological variables, including anti-RgpA and anti-PPAD antibodies, are shown in Table 2. The frequency of periodontitis was similar among the groups; however, the severity was higher for moderate forms in the RA group (*p* < 0.001).

Although *P. gingivalis* presence and quantity did not differ between the patient and control groups, anti-RgpA was significantly higher in RA patients using the Q2 and Q1 grouping (greater than 50th and 75th percentiles, respectively) (Table 2). There was a highly significant association between anti-RgpA and anti-PPAD antibodies (using the quantitative data of dilution 1:400) and RA diagnosis (*p* < 0.001) (Figure 2). However, the anti-PPAD titers adjusted to quartiles did not differentiate between the RA and control groups (Table 2). The anti-RgpA and anti-RgpAQ1–anti-PPADQ2 interactions showed significant differences between the RA and control groups (*p* < 0.001), as did the combination anti-RgpAQ1–anti-PPADQ1 (*p* < 0.05) (Table 2). In contrast, clinical indices such as the number of teeth, IP, PD, CAL, and disease extension (percentage of CAL ≥ 5 mm) were significantly higher in the RA group (*p* < 0.001). The GI and BOP did not differ between the groups (Table 2). A significant association was observed in bivariate analysis between anti-RgpA Q1, BOP CAL, and systemic RA markers such as ACPAs (*p* < 0.05) (Table 3). Interestingly, an association was also observed between the combination of anti-RgpAQ2–anti-PPADQ1 with ACPAs (*p* < 0.05) (Table 3). However, only RgpAQ1 or RgPAQ2 but not anti-PPAD or its statistical interaction showed an association with inflammatory indices of periodontitis in RA individuals (Appendix A). Higher titers of PPAD antibodies were associated with periodontitis and periodontitis severity in RA individuals but not in the activity of RA (Appendix A).

Table 4 shows the different logistic regression models for RA markers, including hsCRP, ESR, RF, ACPAs, anti-RgpA (Model 1), anti-RgpA, anti-PPAD, and their double positivity (Model 3), adjusted for age, sex, BMI, SH, and periodontitis presence. Model 2 was not used because anti-PPAD antibodies did not show any differences between the groups. The calculated OR for these variables were as follows: in Model 1, RF [odds ratio (OR) 10; 95% confidence interval (CI) 4.10–23] and ACPAs (OR 13.7; 95% CI 5.1–36.2). Higher anti-RgpA (Q1) levels were associated with RA (OR 4.09; 95% CI 1.20–13.9). The values calculated in Model 3 were RF (OR 9.9; CI 4.4–25.4), ACPAs (OR 13.7; 95% CI 5.1–36.2), and a combination of anti-RgpA/anti-PPAD double positive (OR 6.63; 95% CI 1.61–27) in association with RA. Next, the OR calculations were modified for Model 1 to the anti-RgpA–anti-PPAD double positive, and an additive interaction between these variables was observed (Table 4).

The operational abilities of anti-RgpA, anti-PPAD, and anti-RgpA/anti-PPAD combinations used as part of a diagnostic test are shown in Appendix A. Anti-RgpA showed a specificity of 87.4% and a PPV of 76.7% for RA diagnosis. However, the anti-RgpAQ1–anti-PPAD-Q2 interaction improved the specificity to 93.7% and PPV to 82.5% for identifying individuals with RA (*p* < 0.05) (Appendix A).

## 4. Discussion

Different clinical and epidemiological studies have corroborated the bidirectional association between RA and periodontitis [32,33,34,35,36] and confirmed that periodontal treatment in patients with RA decreases disease activity [37,38]. This association is attributed to *P. gingivalis* acting as a protagonist, mainly due to the presence of its PPAD, which is linked to the disruption of immunological tolerance in RA [18,39,40]. Although it is currently known that other *Porphyromonas* species (*Porphyromonas gulae* and *Porphyromonas loveana*) have PAD enzymes [41], these species have not been associated with RA.

This study evaluated 143 patients with RA and 112 controls to compare RA biomarkers, such as ACPAs, ERS, hsCRP, RF, anti-RgpA, anti-PPAD, and anti-RgpA/anti-PPAD antibodies as possible systemic markers of RA. SH, which is known to be a risk factor for RA [42,43], did not differ between groups, probably due to the low frequency of smokers in the RA groups with at least 2 years of diagnosis. Additionally, BMI, which has also been associated with RA [42,44,45,46], also did not show differences between the groups. However, age was higher in patients with RA, and more men were observed in the control group than in the RA group (12% vs. 26%). This is a limitation of this study since it was challenging to find systemically healthy individuals without other comorbidities with ages and sexes similar to the group of patients with RA. Therefore, models adjusted for age and sex were used to establish associations between RA markers and anti-RgpA and the combination of anti-RgpA/anti-PPAD antibodies.

More than 70% of the participants presented with periodontitis, without significant differences in the clinical indices of periodontitis. This differs from other reports in which patients with RA have a higher frequency of periodontitis than control groups, especially when the duration of RA is >5 years [13]. Interestingly, patients with RA had a more moderate form of periodontitis, affecting a smaller number of teeth. Additionally, PD, CAL, and extension evaluated by the percentage of CAL ≥ 5 mm were higher in the RA group, consistent with previous reports [33,43,47].

Anti-*P. gingivalis* antibodies accompany RA development, supporting a link between RA and periodontitis [14,48]. A meta-analysis of 14 publications that evaluated the detection of anti-RgpB and anti-*P. gingivalis* in 3829 patients with RA and 1239 controls, indicated a positive relationship between anti-*P. gingivalis* and ACPAs [49]. Okada et al. [50] associated the presence of anti-*P. gingivalis* antibodies with ACPAs and RF; however, these antibodies were also associated with local markers of periodontitis, such as PD and CAL in patients with RA. Similar results were observed in our study: anti-RgpA antibodies and anti-RgpA/anti-PPAD antibodies were higher in RA patients. Additionally, anti-RgpA and anti-RgpA/anti-PPAD antibody interactions were correlated with RF and ACPAs. However, elevated anti-RgpA antibodies correlated with markers associated with periodontitis, such as IG and PI, BOP, PD, and CAL. Other cohort studies have not reported differences in anti-*P. gingivalis* antibodies between patients with RA and controls and only reported a weak statistical significance for ACPAs and FR [36,51]. de Smit et al. [52] in a follow-up study of 289 patients at risk of developing RA, reported that although anti-*P. gingivalis* was associated with RA, these were not prognostic factors for the development of the disease. In 2016, our group investigated healthy individuals with early RA and reported an association between anti-*P. gingivalis* IgG2 and ACPAs supporting the epidemiological association between periodontitis and stages before RA [17].

Serum indicators of infection such as antibodies are the best indicators to study these associations; the clinical indices and the presence of *P. gingivalis* can be modulated by anti-inflammatory treatments of RA as supported by previous data from our research group [17,53,54]. In the Colombian population, the prevalence of *P. gingivalis* in patients with RA and a control group was similar to our results [48]. Other authors have reported that anti-*P. gingivalis* is associated with RA and not with the diagnosis of periodontitis. In RA autoimmunity, antibodies against citrullinated proteins play a very important role and it could be assumed that patients with periodontitis are exposed to citrullinated antigens of *P. gingivalis* that would be systemic immunogens, but antibody titers are higher in RA patients [55,56,57]. *P. gingivalis* is high even in periodontally healthy individuals [58]. In RA, not necessarily high titers of antibodies against anti-*P. gingivalis* its virulence factors must present only in patients with periodontitis since the antigenic challenge of the bacteria could be sufficient to generate the systemic response in these patients. *P. gingivalis* has an aberrant capacity to produce citrullinated peptides through its PPAD and gingipains, favoring the loss of immunological tolerance in patients with risk factors associated with RA [59]. A systematic review concluded that exposure to *P. gingivalis* was a potential risk factor in RA and that anti-P. gingivalis, anti-RgpA, and anti-RgpB are biomarkers associated with RA [60].

Previous studies evaluated the role of gingipains in the citrullination process mediated by PPAD [16,18,52] and the impact of purified anti-RgpB antibodies in this process [61]. Quirke et al. [18] observed in a cohort study that anti-RgpB antibodies were not associated with patients with RA and that only PPAD could be responsible for the loss of immunological tolerance in RA. However, in other studies, anti-RgpB levels were significantly increased in pre-symptomatic individuals and were detectable years before the onset of symptoms of RA [16] and established RA [62].

This is the first study to evaluate the humoral response to native RgpA in patients with RA. We assessed the association of anti-RgpA and anti-PPAD antibodies, and the interaction of anti-RgpA/anti-PPAD antibodies in individuals with RA. We found that patients with high titers of anti-RgpA were associated with RA diagnosis, with an OR of 4.09 (95% CI 1.20–13.9). Additionally, a stronger association between RA and double-positive anti-RgpA/anti-PPAD antibodies was also observed, with an OR of 6.63 (95% CI 1.61–27), suggesting an additive association between these two markers. However, anti-PPAD antibody values did not show an association with RA during the logistic regression analysis.

These results support the hypothesis that the presence of anti-RgpA antibodies may potentiate anti-PPAD antibodies in RA. The link between RA and periodontitis is associated with the citrullinating capacity of exogenous PPAD, generating citrullinated neoepitopes recognized by the immune system, leading to an increase in ACPAs in the host. However, PPAD can only citrullinate arginine at the C-terminal end, and there the RgpA-gingipains play a fundamental role, exposing this arginine that is then targeted for citrullination [18]. Other authors [20,21,63] have found similar results, and they have not found differences these antibodies between RA patients and controls or pre-AR and controls. However, the authors did not evaluate the interaction with antibodies against other antigens such as *P. gingivalis* gingipains. The results of this study provide indirect evidence supporting this concept, in which both bacterial proteins circulate in patients and work collaboratively, and Arg-gingipains enhance the pathogenicity of PPAD.

Authors such as Muñoz-Atienza et al. [63] have indicated that the citrullinated *P. gingivalis* proteome does not contain strongly immunogenic epitopes targeted by serum ACPA during the early stages of RA in this group of patients with established RA with more than 2 years of evolution, as no significant association was found between the diagnosis of RA with high anti-PPAD titers, which agrees with our results. As several authors have discussed, it is not clear whether antibodies against anti-*P. gingivalis* or its virulence factors such as PPAD and gingipains are a consequence or cause of autoimmunity. In this work, it was only possible to show that there were statistical associations with the double positivity of anti-RgpA/anti-PPAD in patients with RA, without being able to confirm whether these are a cause or a consequence of the autoimmunity of the patients and also at what moment they are detectable in patients with RA; this is a limitation in the understanding of the causal association between the presence of *P. gingivalis* in patients with RA.

Kharlamova et al. [62] suggested that anti-RgpB production is not a cause but a consequence of RA and that *P. gingivalis* is a candidate to trigger or drive autoimmunity in patients with RA. The evidence presented here quantifying these antibodies and their double positivity indicate that they could be important association markers to define or confirm RA. The anti-RgpA titer showed high specificity but low sensitivity in identifying patients with RA. However, the double positivity between the two antibody titers improved the diagnostic ability, improving the specificity to 93.7% and, therefore, the PPV. Consequently, these markers must be evaluated to identify RA and its progression. Despite the findings and statistical associations found in this study, the detection of anti-RgpA antibodies and the simultaneous detection of anti-RgpA/anti-PPAD antibodies could not replace or equal the current diagnostic capacity of RA.

This study included individuals with established RA with more than 2 years of evolution. Additionally, they had a high prevalence of periodontitis (83.2%) and *P. gingivalis* infection (46.9%). Therefore, titers of antibodies against this pathogen were high in the population studied. It is likely that the longer the time of diagnosis allows for a longer exposure time to the bacterial antigen, inducing the higher the titers of antibodies against the antigen, and thus patients with RA have higher levels of antibody production.

South America has been considered as the part of the American continent with the highest prevalence and severity of periodontal disease [64]. Periodontitis in patients with RA and controls without RA were similar. In our population, the frequency of moderate periodontitis was high (61.6%) in patients for the age of the patients included in this study [65]. However, the antibodies against different structures of the bacteria were a systemic marker and was only related to the diagnosis of RA.

Eriksson et al. [33] demonstrated that periodontitis and RA increase the expression of APRIL (a proliferation-inducing ligand), a member of the TNF-receptor superfamily, in serum and saliva. This cytokine is essential for B-lymphocyte proliferation, survival, and maturation. This could be involved in the association between RA and periodontitis, based on increased antibody titers. Our study showed that high titers of anti-RgpA/anti-PPAD could be used as markers of RA as a rapid diagnostic test for screening in the future, with potentially better sensitivity than tests using APRIL.

Selection of the individuals in the control group was quite tricky and challenging to match with RA individuals. The incidence of periodontitis in Colombia is very high, and a comparison group free of periodontitis was impossible. However, the results were evaluated by different analyses and are conclusive.

## 5. Conclusions

Anti-RgpA antibodies potentiate the association of anti-PPAD with RA. Anti-PPAD and anti-RgpA–anti-PPAD double positivity can be used to screen RA associated with periodontal disease. Future studies should investigate these antibodies as surrogate markers for the presence and activity of RA.

## Figures and Tables

**Figure 1 jcm-12-03027-f001:**
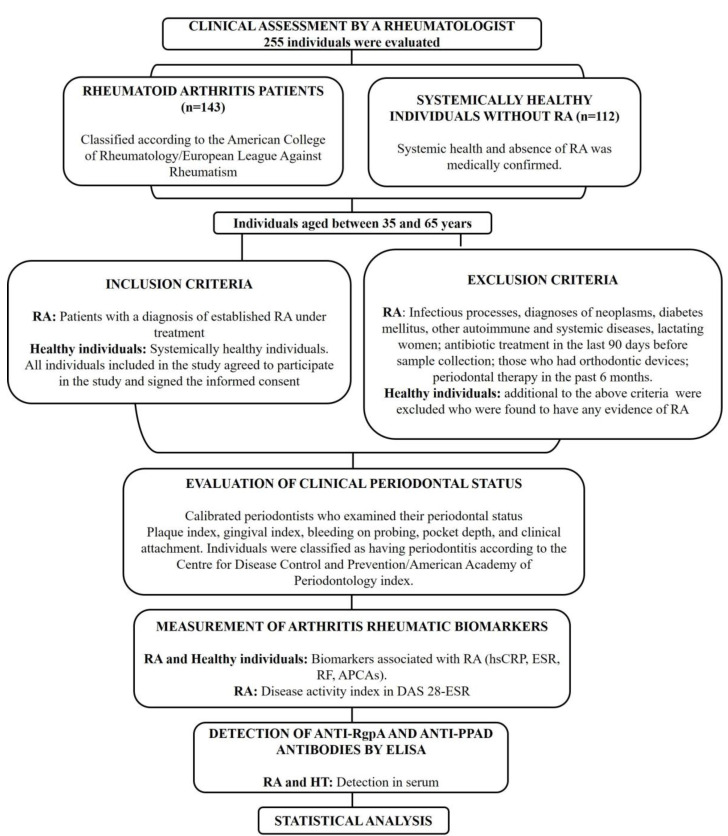
Flow chart of study population selection.

**Figure 2 jcm-12-03027-f002:**
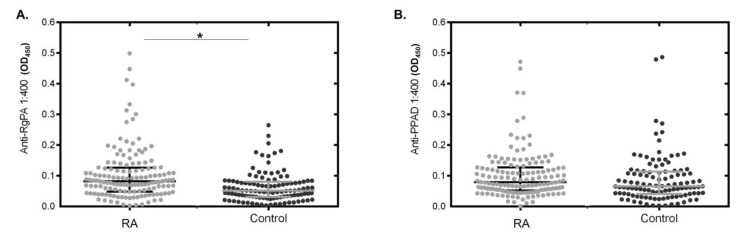
(**A**). Box plot of anti-RgpA antibody concentration in RA and control groups. (**B**). Box plot of anti-PPAD concentration in RA and control groups. * *p* < 0.05.

**Table 1 jcm-12-03027-t001:** Description of socio-demographic and clinical variables and markers associated with RA.

Variable	RA *n* = 143	Control *n* = 112	*p*-Value
**Age**			**<0.0001**
Median (IQR)	57 (51–61)	46 (39–53)	
**Gender F (%)**			**0.039**
Female	125 (87.4)	87 (77.7)	
Male	18 (12.6)	25 (22.3)	
**BMI F (%)**			0.230
Normal	73 (51.0)	66 (58.9)	
Overweight	53 (37.5)	39 (34.8)	
Obesity	17 (11.9)	7 (6.3)	
**Habit smoker F (%)**			0.330
Yes	5 (3.5)	7 (6.3)	
No	138 (96.5)	105 (93.7)	
**ACPAs F (%)**			**<0.0001**
<20 U	25 (17.5)	106 (95.5)	
20–40 U	5 (3.5)	5 (4.5)	
>40 U	113(79)	0 (0)	
**ESR F (%)**			**0.004**
<20 mm/h	80 (55.9)	82 (73.2)	
≥20 mm/h	63 (44.1)	30 (26.8)	
**hsCRP F (%)**			**<0.0001**
Normal	23 (31.1)	44 (59.5)	
>3 mg/L	17 (23)	20 (27.0)	
>10 mg/L	34 (46)	10 (13.5)	
**RF F (%)**			**<0.0001**
<20 IU/mL	19 (25.7)	68 (91.9)	
20–59 IU/mL	19 (25.7)	6 (8.1)	
>59 IU/mL	36 (48.6)	0 (0)	
**DAS28-hsCRP**			
Median RIQ	2.65 (2.2–3.4)	NA	
**RA activity F (%)**		NA	
Without activity	71 (46.65)		
Low activity	25 (17.48)		
Moderate activity	41 (28.67)		
High activity	6 (4.19)		
**Medication F (%)**		112 (100)	**<0.0001**
None	2 (1.4)		
Conventional	110 (76.9)		
Biological	31 (21.6)		
**Diagnosis time of RA F (%)**		NA	
2–5 years	10 (7)		
5–10 years	64 (45)		
>10 years	69 (48)		

BMI = Body Mass Index; ACPAs = IgA antibodies against anti-peptide cyclic citrulline; ESR = Erythrocyte Sedimentation Rate Test; hsCRP = high sensitivity C Reactive Protein; RF = rheumatoid factor. Statistics significance *p* < 0.05. Bolded *p*-values indicate statistically significant results. RA activity = DAS 28: Low (2.43–4.04), middle (4.05–6.31), high (6.32–8.3), very high (≥8.4). NA = Not available.

**Table 2 jcm-12-03027-t002:** Description of periodontal clinical and microbiological variables associated with RA.

Variable	RA *n* = 143	Control *n* = 112	*p*-Value
**Periodontitis** F (%)			0.106
Presence	119 (83.2)	84 (75)	
Absence	24 (16.8)	28 (25)	
**Severity** F (%)			**0.001**
None	24 (16.8)	28 (25)	
Low	1 (0.7)	18 (16.1)	
Mild	105 (73.4)	54 (48.2)	
Severe	13 (9.1)	12 (10.7)	
**Number of Teeth**			**<0.0001**
Median	20	25	
(IQR)	(13–25)	(21–28)	
**Plaque Index%**			0.023
Median	68	52.5	
(IQR)	(35–85)	(35–69)	
**Gingival Index%**			0.117
Median	50	32	
(IQR)	(10–73)	(17.5–63)	
**Bleeding on probing (%)**			0.381
Median	47	40	
(IQR)	(27–63)	(27–54)	
**Pocket Depth (mm)**			**<0.0001**
Median	4	4	
(IQR)	(0–4.2)	(4–4.3)	
**Pocket Depth (%)**			0.342
Median	25.3	24.54	
(IQR)	(23–40)	(10–39.7)	
**CAL (mm)**			**0.0002**
Median	2.74	2.38	
(IQR)	(2.26–3.39)	(1.94–2.95)	
**CAL > 5 mm (%)**			**<0.0001**
Median	87	51.6	
(IQR)	(60.7–97.7)	(34.6–77.1)	
***P. gingivalis*** F (%)			0.287
Presence	67 (46.9)	60 (53.6)	
Absence	76 (53.1)	52 (46.4)	
***P. gingivalis* Log_10_**			0.491
Median	0	4.14	
(IQR)	(0–6.3)	(0–6.3)	
**Anti-RgpA Q1** F (%)			**<0.0001**
Positive +	88 (61.5)	39 (34.8)	
Negative −	55 (38.4)	73 (65.2)	
**Anti-RgpA Q2** F (%)			**<0.0001**
Positive +	45 (31.5)	15 (13.4)	
Negative −	98 (68.5)	97 (86.4)	
**Anti-PPAD Q1** F (%)			0.059
Positive +	84 (59)	53 (42)	
Negative −	59 (41)	59 (58)	
**Anti-PPAD Q2** F (%)			0.236
Positive +	54 (37.7)	35 (32.3)	
Negative −	89 (62.3)	77 (77.7)	
**Anti-RgpAQ1-PPADQ1**			**0.026**
Positive +	33 (23.1)	8 (7.2)	
Negative −	110 (76.9)	104 (92.8)	
**Anti-RgpAQ1-PPADQ2**			**<0.0001**
Positive +	18 (12.5)	6 (5)	
Negative −	125 (87.4)	106 (95)	

Anti-RgpA = anti-Arg-gingipain antibodies; PPAD = peptidyl arginine deiminase of *P. gingivalis*; CAL = clinical attachment loss. % = expressed in percentages; mm = expressed in millimeter; Statistical significance *p* < 0.05. Bolded *p*-values indicate statistically significant results.

**Table 3 jcm-12-03027-t003:** Comparison of anti-RgpA, anti-PPAD, and anti-PPAD-RgpA interaction with RA markers (A) and periodontitis markers in the entire population (B).

Variable	Anti-RgpA Q2		Anti-RgpA Q1		Anti-PPAD Q2		Anti-PPAD Q1		Anti-PPADQ2RgpA Q1	
	Positive	Negative	Positive	Negative	Positive	Negative	Positive	Negative	Positive	Negative
**A. RA markers**										
**ACPAs** F (%)										
<20 U	51 (39)	80 (61)	23 (18)	108(82)	64 (49)	67 (51)	39 (30)	92 (70)	14 (35)	117 (55)
20–40 U	4 (40)	6 (60)	2 (20)	8 (80)	4 (40)	6 (60)	2 (20)	8 (80)	1 (25)	9 (4)
>40 U	71 (63) †	42 (37)	35 (31)	78 (69)	68 (69)	45 (31)	47 (42)	66 (58)	25 (62.5) **	88 (41)
**RF** F (%)										
<20 IU/mL	46 (35)	84 (65)	21 (16)	109 (84)	63 (48)	67 (52)	39 (30)	91 (70)	14 (35)	116 (54)
20–59 IU/mL	28 (68)	13 (32)	16 (39)	25 (61)	26 (53)	15 (37)	15 (37)	26 (63)	10 (25)	31 (15)
>59 IU/mL	52 (63) **	31 (37)	23 (28)	60 (72)	48 (58)	36 (42)	34 (41)	49 (59)	16 (40)	67 (31)
**B. Periodontal markers**										
**Gingival Index**										
Median	47	35	46	36	38	37	37	38	35	38
IQR	(13–68)	(18–63)	(8–65)	(18–64)	(14–66)	(15–66)	(14–64)	(15–64)	(6–64)	(17–65)
**BoP**										
Median	47 **	40	52	40	43	41	44	41	41	47
IQR	(27–64)	(23–55)	(23–67)	(26–56)	(22–60)	(27–59)	(21–59)	(26–59)	(26–58)	(20–63)
**Pocket depth**										
Median	4	4	4	4	4	4	4	4	4	4
IQR	(0–4.2)	(0–4.19)	(0–4.0)	(0–4.16)	(0–4.25)	(0–4.17)	(1–4.38)	(0–2.5)	(0–4.14)	(0–4.0)
**CAL**										
Median	2.75 **	2.48	2.79 **	2.52	2.7	2.6	2.6	2.7	2.6	2.8
IQR	(2.3–3.3)	(2–2.9)	(2.1–3.6)	(2–3)	(2.1–3.2)	(2.1–3.2)	(2.1–3.2)	(2.2–3.2)	(2.1–3.1)	(2.3–3.4)

† *p* < 0.001; ** *p* < 0.05. RgpA = arginine-gingipain; PPAD = *P. gingivalis*-peptidyl arginine deiminase; ACPAs = IgG/IgA antibodies against anti-peptide cyclic citrulline; RF = rheumatoid factor. BoP = bleeding on probing; CAL = clinical attachment loss.

**Table 4 jcm-12-03027-t004:** Regression logistic model of the markers associated with RA.

	Reference	OR Unadjusted	(CI 95%)	OR Adjusted	(CI 95%)
**Model 1** **Dependent Variable**					
**RA**	Without RA				
**Independent variable**					
ACPAs < 20 U	Reference	1.0		1.0	
ACPAs > 20 U		13.7	5.1–36.2 **	15.21	5.2–44.5 **
RF < 20 UI/mL	Reference	1.0		1.0	
RF > 20 UI/mL		10.6	4.10–23 **	15.58	5.0–48.49 **
Anti-RgpA < Q1	Reference	1.0		1.0	
Anti-RgpA > Q1		4.09	1.20–13.9 *	8.6	1.56–47.8 **
**Model 3** **Dependent Variable**					
**RA**	Without RA				
**Independent variable**					
ACPAs < 20 U	Reference	1.0		1.0	
ACPAs > 20 U		13.7	5.1–36.2 **	14.4	5.2–39.9 **
RF < 20 UI/mL	Reference	1.0		1.0	
RF > 20 UI/mL		9.9	4.4–25.4 **	14.6	4.7–45 **
Anti-RgpA < Q1Anti-PPAD < Q2	Reference	1.0		1.0	
Anti-RgpA > Q1Anti-PPAD > Q2		6.63	1.61–27 *	9.69	1.77–52 *

ACPAs = antibodies to citrullinated protein antigens; RF = rheumatoid factor; Anti-RgpA = Arg-gingipains antibodies; Anti-PPAD = *P. gingivalis* peptidyl arginine deiminase antibodies; Q1 ≥ 75th percentile; Q2 ≥ 50th percentile. Model adjusted to age, gender, BMI, smoking, and periodontitis presence. ** *p* < 0.001: * *p* < 0.05. **Model 1**. LR: *p* = 0.25; BIC: unadjusted = 119.15; adjusted = 127.49, difference = −8.34; AIC: unadjusted = 101.46; adjusted = 102.7, difference = −1.24. The **unadjusted model** should be reported. **Model 3.** LR: *p* = 0.45; BIC: unadjusted = 117.2; adjusted = 126.7 difference = −9.5; AIC: unadjusted = 99.5; adjusted = 101.9, difference = −2.4. The **unadjusted model** should be reported. Age was a confounder variable.

## Data Availability

All data needed to evaluate the conclusions of the paper are presented in the paper. Additional data related to this paper may be requested from the authors.

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
