# Peer review of "The Interaction Effect of Anti-RgpA and Anti-PPAD Antibody Titers: An Indicator for Rheumatoid Arthritis Diagnosis"

_jcm, 2023, doi:10.3390/jcm12083027_

Round 1

Reviewer 1 Report

This is an interesting paper linking, for the first time, RgpA antibodies to risk of RA as well as correlating these with presence of PPAD antibodies as a co-modulator of risk. The methods and analysis are well performed. I have only one concern which is that, despite the statistical significance of the results, the actual percent of RA patients identified either by RgpA or RgpA/PPAD antibodies is actually very small. I think that it is incumbent on the authors to add a paragraph discussing the limitations of the results. In particular, a standard way of looking at the question would be how many patients would need to be tested in order to diagnose RA using the new tests as compared with existing diagnostic methods. In other words, how much do the new correlations actually add to our diagnostic ability?

Another aspect of the limitations discussion should more carefully examine the possibility that the results do not indicate that P. gingivalis is a cause/trigger for RA but rather a target of RA autoantibodies that cross-react with P. gingivalis proteins.  It is well-established for a wide range of autoimmune diseases that significant alterations in the host microbiome occur in conjunction with the disease but it is not known whether the attack on the microbiome is a consequence or a cause of the autoimmunity.  Thus, it is important for the authors to discuss whether the limitations of their study with regard to this issue, in particular in light of the fact that all of their patients had RA for more than 2 years. This point is particularly important in light of the following study, which although cited by the authors, is not discussed in the context of WHEN the antibodies appear during the disease course: Muñoz-Atienza E, Flak MB, Sirr J, Paramonov NA, Aduse-Opoku J, Pitzalis C, Curtis MA. The P. gingivalis Autocitrullinome Is Not a Target for ACPA in Early Rheumatoid Arthritis. J Dent Res. 2020 Apr;99(4):456-462. doi: 10.1177/0022034519898144.

One additional concern is that since RgpB antibody titers are already correlated with RA, and because RgpA and RgpB are significantly homologous, it would have been appropriate for the authors to have tested for RgpB antibody as well as RgpA antibody to determine to what extent determining RgpA antibody titers are an improvement on RgpB titers.  I presume that this type of measurement is not now possible, but if the authors happen to have RgpB titers for their patients, or could easily determine them, this would vastly improve the quality and importance of this paper.

Author Response

Dear Evaluator

We greatly appreciate your comments and below they are answered point by point. 

Reviewer 1

  1. This is an interesting paper linking, for the first time, RgpA antibodies to risk of RA as well as correlating these with presence of PPAD antibodies as a co-modulator of risk. The methods and analysis are well performed. I have only one concern which is that, despite the statistical significance of the results, the actual percent of RA patients identified either by RgpA or RgpA/PPAD antibodies is actually very small. I think that it is incumbent on the authors to add a paragraph discussing the limitations of the results.

Answer: Thank you very much for your appreciation and for showing the importance of this work. Added a paragraph with the limitations of this study in the discussion.

  1. In particular, a standard way of looking at the question would be how many patients would need to be tested in order to diagnose RA using the new tests as compared with existing diagnostic methods. In other words, how much do the new correlations actually add to our diagnostic ability?

Answer: Your observation is very interesting, as you confirm, the frequency of anti-RgpA detection and the simultaneous detection of anti-RgpA/anti-PPAD is not as high as would be expected in the RA group. Therefore, it is clarified in the text that the detection of these antibodies is not intended to replace the conventional methods of diagnosis of RA (lines 447 to 450), but would be a screening that in the future is intended to help the clinician to screen patients with systemic risk to RA.

Additionally, as indicated in table S3, the sensitivity is not very high, but the specificity is, in that order of ideas we could not answer your question how many patients would need to be tested in order to diagnose RA using the new tests as compared with existing diagnostic methods.

  1. Another aspect of the limitations discussion should more carefully examine the possibility that the results do not indicate that gingivalis is a cause/trigger for RA but rather a target of RA autoantibodies that cross-react with P. gingivalis proteins.  It is well-established for a wide range of autoimmune diseases that significant alterations in the host microbiome occur in conjunction with the disease, but it is not known whether the attack on the microbiome is a consequence or a cause of the autoimmunity.  Thus, it is important for the authors to discuss whether the limitations of their study with regard to this issue, in particular in light of the fact that all of their patients had RA for more than 2 years. This point is particularly important in light of the following study, which although cited by the authors, is not discussed in the context of WHEN the antibodies appear during the disease course: Muñoz-Atienza E, Flak MB, Sirr J, Paramonov NA, Aduse-Opoku J, Pitzalis C, Curtis MA. The P. gingivalisAutocitrullinome Is Not a Target for ACPA in Early Rheumatoid Arthritis. J Dent Res. 2020 Apr;99(4):456-462. doi: 10.1177/0022034519898144.

Answer: Thank you very much for your observation, as indicated is a limitation of this study and a discussion paragraph is included in this regard on lines 428 to 439

  1. One additional concern is that since RgpB antibody titers are already correlated with RA, and because RgpA and RgpB are significantly homologous, it would have been appropriate for the authors to have tested for RgpB antibody as well as RgpA antibody to determine to what extent determining RgpA antibody titers are an improvement on RgpB titers. I presume that this type of measurement is not now possible, but if the authors happen to have RgpB titers for their patients, or could easily determine them, this would vastly improve the quality and importance of this paper.

Answer: Thank you very much for your suggestion, as you indicate, RgpB and RgpA present a 98% homology in their catalytic domain, and differ in that RgpA presents the hemagglutinin adhesin domain, due to the previously described methodological strategy (Castillo DM, Castillo Y, Delgadillo NA, et al. Purification of RgpA from external outer membrane vesicles of Porphyromonas gingivalis. Anaerobe. 2022;77:102647. doi:10.1016/j.anaerobe.2022.102647), the protein used as antigen for the detection of anti-RgpA antibodies of this work includes both domains, so we believe that due to the homology of the catalytic domain, anti-RgpB antibodies could be very similar in this group of patients or even inferior because they do not have the hemagglutinin adhesin domain. However, as you indicate, it would be very interesting to be able to compare the anti-RgpA antibody titers with the anti-RgpB antibodies in the next study.

Attached the document with the corrections highlighted in yellow

Sincerely,

DIANA MARCELA CASTILLO

Reviewer 2 Report

Recommended

Change the title to "Interaction of anti-RgpA and anti-PPAD antibody titers: an index in the diagnosis of rheumatoid arthritis"

Edit the sentences grammatically for better understanding.

and employ the articles like DOI: 10.18502/ijaai.v17i4.95

Author Response

Dear Evaluator 

We appreciate all your comments on our work, below we respond point by point observations.

Change the title to "Interaction of anti-RgpA and anti-PPAD antibody titers: an index in the diagnosis of rheumatoid arthritis"

Answer: We appreciate your suggestion; the title is changed according to the evaluator's request.

Figure 1.  What does the FR stand for?

Answer: Thank you very much for your observation, the correction was adjusted

This result is considered positive for both genders, while the ESR test is differently interpreted for both genders. so, this issue needs more explanations in detail.

Answer: Thanks for the observation, it was decided to take the value of 20mm according to the manufacturer's instructions for the general population since no significant results were found and, for the compound activity indices the variable was taken continuously as indicated by the validated formula for the calculation the DAS28.

Considering that only the results of one of the above titles have been presented, therefore, there is no need to work on three titers. It is recommended to read and refer to the article DOI: 10.18502/ijaai.v17i4.95.

Answer: Thank you very much for your observation, we have revised the article and it is corrected, leaving only the dilution in which the results are shown.

Attached the document with the corrections highlighted in yellow

Sincerely,

DIANA MARCELA CASTILLO
